# VISION-BASED PSEUDO-TACTILE INFORMATION EXTRACTION AND LOCALIZATION FOR DEXTEROUS GRASPING

## ABSTRACT

Due to the challenges in acquiring tactile perception during mechanical dexterous hand grasping and the complexity of multi-finger contact, robotic dexterous hand grasping remains a difficult problem. This study addresses two main tasks: (1) Acquiring tactile information from everyday objects using vision, termed "pseudo-tactile" information, and (2) Building a Dexterous Hand (RH8D) model in Isaac Sim for real-time fingertip contact localization. Utilizing Isaac Sim allows for safe, cost-effective experimentation and high-precision simulations, facilitating data collection for model validation. The research establishes a scientific connection between simulated 3D coordinates, actual 3D coordinates, and pseudo-tactile information derived from point clouds, quantified through normal vectors and grayscale variance analysis. Experimental results demonstrate the ability to extract clear object surface textures, accurately locate fingertip contact points in real-time (with precision up to 0.001 m), and provide tactile information at contact points. This framework enhances robotic grasping capabilities and offers low-cost sensory data. The source code and dataset are publicly available at https://github.com/Fenbid0605/Vision-Based-Tactile-Information-Extraction-and-Localization-for-Dexterous-Grasping.

## 1 INTRODUCTION

Dexterous manipulation remains a cutting-edge research direction in robotics, posing challenges for both hardware performance and algorithm development. This study introduces a novel method for vision-based tactile information extraction and contact point localization.

In machine vision, an object's surface tactile information is reflected through its texture, and three-dimensional point clouds, compared to two-dimensional images, better capture surface characteristics. Using an Intel RealSense camera Lourenço & Araujo (2021), we generate point cloud data

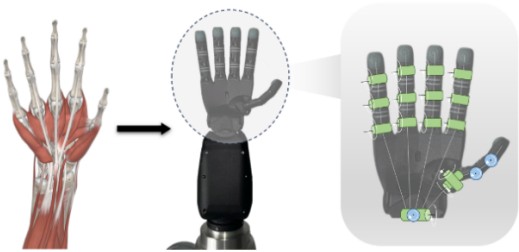

Figure 1: RH8D Adult Robot Hand: 1) Inspired by the human hand, capable of performing critical grips; 2) 19 Degrees of freedom (DoF).

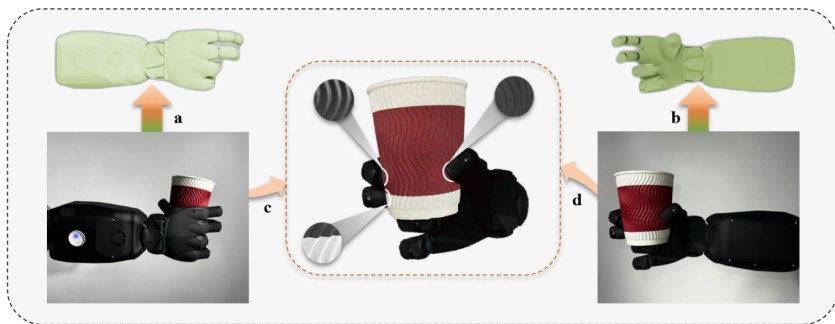

Figure 2: By real-time linking the degrees of freedom of each joint with actual dexterous hand movements and the simulation platform, the RH8D grasping actions are precisely extracted. a, b: From reality to simulation; c, d: Enlarging the clear 2D tactile information at the dexterous hand grasping contact fingertips.

and extract textures by analyzing surface normal vectors and grayscale variance. To address the challenge of localizing contact points during dexterous hand grasping, the study simulates various illumination angles and object materials in Isaac Sim, providing real-time fingertip contact coordinates.

The Seed Robotics RH8D hand, shown in Figure 1, is the primary focus of this study. The RH8D, inspired by the human hand, features an opposable thumb and a spherical wrist joint. By analyzing the grasping of objects with different types, sizes, materials, and weights, we focus on extracting tactile information through vision and accurately localizing fingertip contact points. The point cloud data is processed using a filtering algorithm that calculates normal vectors and grayscale variance. Points, where both the Y-component of the normal vector and grayscale variance exceed a predefined threshold, are classified as texture feature points Kim et al. (2018).

By transmitting the real-time grasping joint states of the dexterous hand to the Isaac Sim platform, the hand's grasping actions are accurately replicated and analyzed, as shown in Figure 2. Utilizing the platform's high-precision simulations, fingertip contact point coordinates are calculated with the wrist as the origin and transmitted to real-world applications, determining the RH8D dexterous hand's spatial contact coordinates. This approach enables the extraction of detailed grasping perceptual information. Thus, this study introduces an innovative method for vision-based tactile information extraction and precise localization.

## 2 RELATED WORK

### 2.1 VISION-BASED GRASPING AND POSITIONING

In recent years, the integration of machine vision and sensing technologies has significantly enhanced robotic hands' environmental perception capabilities. The positioning and grasping technologies of robotic hands are advancing toward more intelligent and adaptive directions. For instance, Teulière & Marchand (2012) used RGB-D data for visual servoing, while Endres et al. (2012) and Whelan et al. (2015) extended SLAM technology to RGB-D data. These integrated perception systems enable robotic hands to more accurately recognize and manipulate objects in complex or dynamically changing environments.

In the field of deep learning, Bai et al. (2022) introduced Spatial-Aware Tokens, a method based on the self-attention mechanism to capture long-distance feature dependencies in images, significantly improving the accuracy of object positioning. Similarly, Flintoff et al. (2018) proposed the Single-Grasp system, which analyzes RGB-D images through neural networks to predict object locations and optimal grasping points.

Building on previous studies, we introduce the concept of "pseudo-tactile" information—tactile-like data derived from visual analysis of an object's surface, in contrast to traditional tactile sensing that relies on physical interaction. This distinction is crucial for recognizing the strengths and limitations

of our approach, as the surface textures we extract are visual representations rather than direct tactile feedback.

## 2.2 DEXTEROUS HAND PERCEPTION

After years of development, robotic perception systems for positioning and tracking have become highly mature Bullock et al. (2013) . With continuous advancements in robotics Bicchi (2000), perception capabilities have become key to achieving high precision and efficiency in robotic operations Ujitoko & Ban (2021). Robotic manipulation technology has evolved from simple mechanical models to complex sensing and control paradigms. Modern methods prioritize real-time perception and dynamic adjustment, where tactile sensors play a crucial role Zapata-Impata et al. (2021) For example, Zapata-Impata et al. (2021) used 3D vision to generate tactile data, enabling real-time mechanical perception during robotic grasping, such as force and friction.

With the development of humanoid robots, dexterous hands have emerged as essential tools for complex operations, requiring highly flexible grasping capabilities and precise positioning Yamaguchi et al. (2013). However, the grasping and positioning of dexterous hands still face significant limitations, such as reliance on high-precision LiDAR or expensive external sensors Ratnasingam & McGinnity (2011). Moreover, existing tactile sensing systems for dexterous hands are generally costly Michelman (1998), and the diverse working principles of these sensors greatly limit their widespread application Hachisu et al. (2011). Therefore, researching a low-cost, high-performance tactile sensing and grasping positioning system for dexterous hands is the primary goal of this study.

## 2.3 "PSEUDO-TACTILE" BASED ON VISUAL SENSORS

Pseudo-haptic technology, as explored by Xavier et al. (2024), simulates haptic feedback by leveraging the brain's processing of tactile and kinesthetic inputs, creating the illusion of tactile signals. This approach allows users to perceive shapes, textures, and other physical properties in virtual environments without physical contact. Xavier et al. (2024) also proposed a new taxonomy for pseudo-haptics, combining multiple feedback mechanisms not addressed in prior research and presenting a multimodal strategy with potential applications in various fields.

Similarly, Sato et al. (2020) introduced a pseudo-haptic feedback framework that provides tactile sensations of objects without physical haptic devices. Their framework visualizes bumpy, slippery, and soft sensations and modulates the intensity of these sensations without inducing unnatural experiences. However, despite these advancements, their system's effectiveness can be influenced by external factors such as ambient light and projection quality, limiting its robustness.

In contrast, our approach of deriving tactile feature information from point cloud images effectively mitigates these limitations, particularly by reducing sensitivity to lighting variations. Additionally, Ujitoko & Ban (2021) synthesized prior research to provide a comprehensive overview of pseudo-haptics, highlighting its potential applications in training, assistive technologies, and entertainment.

Our "pseudo-tactile" technique fundamentally differs from traditional pseudo-haptic methods. Instead of relying solely on visual cues, we use Intel RealSense cameras to capture detailed point cloud data, from which we extract surface undulations to simulate tactile textures. This approach generates "pseudo-tactile" information that accurately reflects the object's physical properties. By precisely locating fingertip contact points and matching them with the corresponding pseudo-tactile data, we enhance tactile feedback reliability in robotic grasping. Unlike existing methods, our approach is less sensitive to lighting and environmental variations, providing more consistent tactile information.

## 3 METHODOLOGY AND ALGORITHMS

### 3.1 GENERATION AND PREPROCESSING OF TACTILE INFORMATION ON OBJECT SURFACES

In machine vision, an object's tactile information is represented by its surface texture, and 3D point clouds more effectively capture these characteristics compared to 2D images. To simulate the generation of surface information through vision, this study employs the Intel RealSense Camera for point cloud acquisition.

### 3.1.1 Robot Vision Acquiring Point Cloud Information

For accurate contact point localization, the object is placed on a horizontal surface with the point cloud acquisition device positioned 60 cm away. The system is calibrated to align the camera's coordinate system with both the workpiece and the global coordinates. Once the object is secured on the platform, data acquisition begins. The robotic arm moves the device around the object, capturing depth and image frames, which are then aligned to generate depth and RGB maps. Using the camera's internal and external parameters (see Table 1), the point cloud is generated within the world coordinate system. A filtering algorithm is applied to identify texture feature points by calculating the normal vector and grayscale variance for each point. Points are classified as texture features if both the Y-component and grayscale variance exceed empirically determined thresholds, optimizing feature identification and minimizing false positives.

Table 1: Camera Internal and External References and Their Meanings

| Intrinsics & Extrinsics | Meaning |
| --- | --- |
| X Coordinate ($PP_x$) | X coordinate of the optical center in the pixel coordinate system |
| Y Coordinate ($PP_y$) | Y coordinate of the optical center in the pixel coordinate system |
| Focal Length in X Direction ($f_x$) | Focal length in the X direction |
| Focal Length in Y Direction ($f_y$) | Focal length in the Y direction |
| Rotation Matrix ($R_{3\times3}$) | Rotation matrix |
| Translation Vector ($T_{3\times1}$) | Translation vector |

*Descriptions of the camera internal and external references.

The entire feature extraction process is visualized in Fig. 4, which presents a flowchart outlining the steps from data acquisition to feature point selection.

According to the transformation relationship between the world coordinate system, the camera coordinate system, and the pixel coordinate system, if we know a point within the pixel coordinate system and the depth d of the point, the following relationships between this point and the point under the world coordinate system can be deduced:

$$d \begin{pmatrix} u \\ v \\ 1 \end{pmatrix} = \begin{pmatrix} f_x & 0 & pp_x \\ 0 & f_y & pp_y \\ 0 & 0 & 1 \end{pmatrix} \begin{pmatrix} x_c \\ y_c \\ z_c \end{pmatrix} \tag{1}$$

$$\begin{pmatrix} x_c \\ y_c \\ z_c \\ 1 \end{pmatrix} = \begin{pmatrix} R_{3\times3} & T_{3\times1} \\ \vec{0}_{1\times3} & 1_{1\times1} \end{pmatrix} \begin{pmatrix} x_w \\ y_w \\ z_w \\ 1 \end{pmatrix} \tag{2}$$

Wherein, the values of the rotation matrix $R_{3\times3}$, and translation vector $T_{3\times1}$ are respectively

$$R_{3\times3} = \begin{pmatrix} 1 & 0 & 0 \\ 0 & -1 & 0 \\ 0 & 0 & -1 \end{pmatrix} \tag{3}$$

$$T_{3\times1} = \begin{pmatrix} 0 \\ 0 \\ 0 \end{pmatrix} \tag{4}$$

In the world coordinate system, the Y-axis serves as the vertical coordinate axis, and the object is positioned vertically, perpendicular to the XOZ plane.

### 3.1.2 Point Cloud Preprocessing

The camera's field of view captures both the object and the surrounding background, such as horizontal and vertical surfaces, resulting in a point cloud that includes both object and background

information. Since the background data does not contribute to the tactile information of the object, we preprocess the point cloud to isolate the object's data within the camera's field of view.

**Coordinate Transformation:** The point cloud visualization indicates that the center point of the generated cloud is located near (0,0,0). The first step in preprocessing is to translate the point cloud so that the minimum values $x_{min}$, $y_{min}$, and $z_{min}$ of the X, Y, and Z coordinates are set to zero. This facilitates the estimation of the upper and lower bounds for pass-through filtering.

**Pass-Through Filtering:** Background point cloud data exists in the X, Y, and Z directions of the world coordinate system. To filter out this background information, we define upper and lower bounds for pass-through filtering in all three directions and retain only the points within these bounds. In this paper, we apply the bisection method to determine the filtering intervals, ensuring the background is removed and the object's point cloud is fully isolated within the camera's view.

### 3.2 Texture Extraction Based on Normal Vectors and Grayscale Value Variance

Due to the discrete nature of the point cloud data acquired by the camera, the points are independent of each other. The texture feature describes the degree of undulation of a certain area of the object's surface, which cannot be characterized by a single point alone. So it is necessary to establish the neighborhood relationship between the points to further extract the texture feature from the point cloudBudiyanta et al. (2021). The flowchart of this tactile feature extraction algorithm is shown in Figure 3.

KD tree is a data structure that is very efficient in neighborhood queries. In this paper, we use the filtered point cloud information to construct and integrate coordinate information and color information in the search process to achieve the extraction of texture features. The algorithm is divided into the following steps:

**KD Tree Construction:** Use the coordinate information of the point cloud $P = \{p_1, p_2, \cdots, p_n\}$ to construct a KD tree, which facilitates efficient neighborhood search.

**Neighborhood Search:** Specify the neighborhood radius, for each point $p_i \in P$, the KD tree will search and return its neighboring point indexes within the radius $r$, the current point $p_i$ and its neighboring points can be accessed via these indexes and these points satisfy:

$$N(p_i, r) = \langle p_j \in P \mid (p_i - p_j)^2 + (p_i - p_j)^2 + (p_i - p_j)^2 \le r \rangle \tag{5}$$

where $p_x, p_y, p_z, p_i$ is the component of the point in X, Y and Z directions respectively.

**Normal Vector Calculation:** The PCA-based method is used to solve the covariance matrix $C$ of the neighboring point set $N(p_i, r)$, and then calculate its eigenvalue and eigenvector. Take the eigenvector corresponding to the smallest eigenvalue of point $p_i$ as the normal vector $n_i$ of it:

$$C = \frac{1}{|N(p_i, r)|} \sum_{p_j \in N(p_i, r)} (p_j - \mu)(p_j - \mu)^T \tag{6}$$

$$n_i = \text{eigenvector}_{\arg\min \lambda}(C) \tag{7}$$

Where $\mu$ is the mean vector of the neighboring point set $N(p_i, r)$.

**Gray Value Variance Calculation:** Calculate the gray value variance $\sigma_i^2$ of the neighboring point set of the current point $p_i$:

$$\sigma_i^2 = \frac{1}{|N(p_i, r)|} \sum_{p_j \in N(p_i, r)} \left( \text{gray}(p_j) - \overline{\text{gray}}(p_j) \right)^2 \tag{8}$$

Where $\text{gray}(p_i)$ is the gray value of point $p_i$ and $\overline{\text{gray}}(p_j)$ is the average gray value of the neighboring point set $N(p_i, r)$.

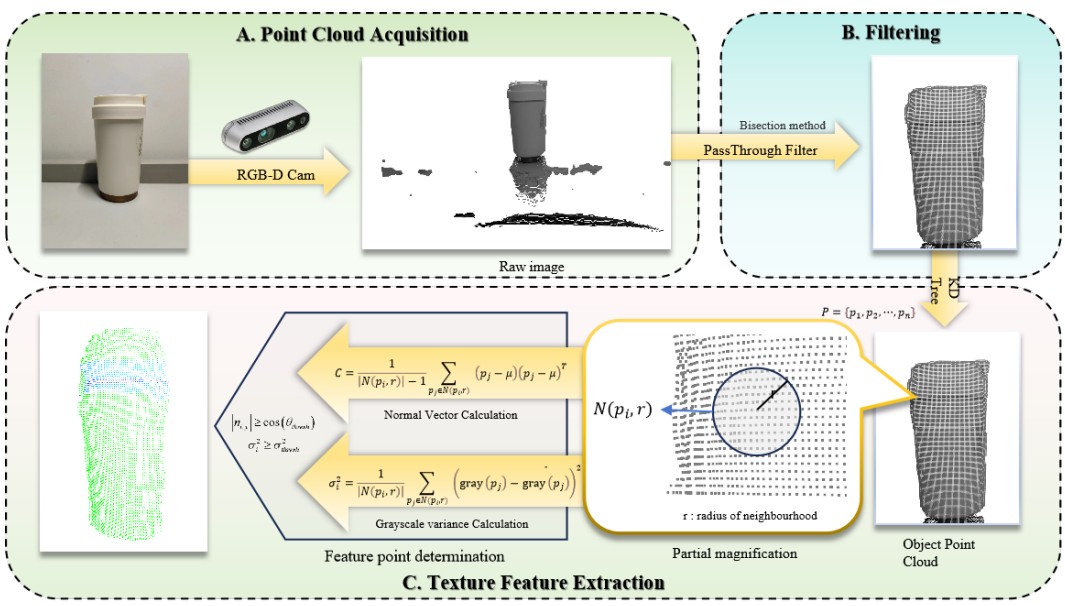

Figure 3: Texture Feature Extraction Flow Chart

**Texture feature point determination:** Specify the threshold $n_y, thresh$ of the $Y$ component of the unit normal vector and the threshold $\sigma^2_{thresh}$ of the variance of the grayscale information, for each point $p_i$, calculate the absolute value $abs(|\vec{n}^0_{i,y}|)$ of the $Y$ component of the unit normal vector of it, and if the following conditions are satisfied, it is considered to be a texture feature point:

$$abs\left(|\vec{n}^0_{i,y}|\right) \geq n^0_{y,\text{thresh}} \qquad (9)$$

$$\sigma^2_i \geq \sigma^2_{thresh} \qquad (10)$$

**Classification and Visualization:** The points that meet the conditions are classified as texture feature points, while the rest are classified as non-texture feature points. The texture feature points are visualized to illustrate the extracted texture features of the point cloud, allowing the dexterous hand to obtain more valid target information and providing more comprehensive perceptual information for the dexterous hand to manipulate precisely.

### 3.3 SPATIALLY ACCURATE LOCALIZATION OF FINGERTIP CONTACT POINTS

In automation and robotics engineering, achieving high-precision grasping is essential for the commercial deployment of dexterous hands. This study employs the NVIDIA Isaac Sim platform for precise spatial localization of dexterous hand contact points, facilitating accurate acquisition of tactile texture data at the contact points. This approach allows for precise control of experimental conditions and parameters, enabling the collection of contact point coordinates and the corresponding degrees of freedom for various grasping postures.

A highly realistic dexterous hand model and corresponding operational environment were constructed to replicate the physical properties and conditions of the real world. Leveraging the advanced physical engines and graphics rendering capabilities of the Isaac Sim platform, we simulate various grasping scenarios with objects of different materials, shapes, and sizes (see Figure 6, top row).

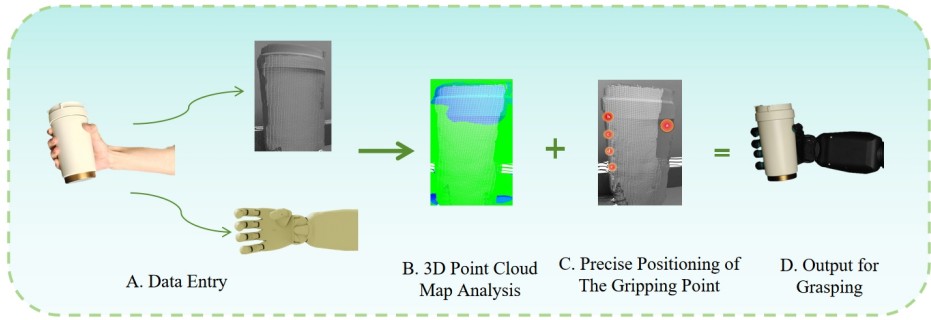

Figure 4: Dexterous hand five-finger grasp between the spatial coordinates of the contact point flowchart: Based on NVIDIA Isaac Sim's dexterous hand grasping, fingertip contact points are spatially accurately located.

To address the contact point localization issue in dexterous hand grasping operations, we simulate various illumination angles and different materials and shapes of objects during the simulation process. We also employ policy fine-tuning techniques, using a small amount of real-world grasping attempt data to fine-tune the model, further reducing the gap between simulation and reality. Through high-precision calculations and simulations based on Isaac Sim, we use the wrist as the coordinate origin and accurately determine the coordinates of the dexterous hand's five fingertip contact points and the corresponding joint angles based on the object's three-dimensional geometric features and surface material (as shown in Figure 4).

Combining the tactile texture information at the fingertip contact points provides finer localization and sensory information for grasping in complex environments, offering low-cost multimodal sensory information for the field of embodied intelligent agents.

## 4 EXPERIMENT

- Provide a detailed description of the dataset, experimental setup, and dexterous hand manipulation process.

- Evaluate whether robot vision alone can capture 3D tactile surface data during multi-finger grasps.

- Assess the ability of our approach to compute real-time fingertip positions for five fingers and validate the results in practical scenarios.

### 4.1 EXPERIMENT SETTINGS

**Dataset:** We collected point cloud images of over 200 everyday objects, encompassing a wide variety of materials, shapes, and surface textures, to build a comprehensive open-source test dataset. Lighting parameters (brightness and color temperature) were adjusted to simulate different conditions, enhancing the algorithm's robustness to light variations. Additionally, we conducted grasping operations with a robotic hand to gather real grasping data, including fingertip contact points coordinates and joint angles at various time points. To ensure broad applicability, the dataset includes diverse objects (e.g., glass, plastic) and shapes (e.g., round, square) for training and validation. From these, 49 representative objects were selected for deep learning experiments, and 15 were used for final validation. To promote scientific transparency and reproducibility, the public dataset is available (see Appendix A.2 for details).

**Experimental Procedure:** In this study, we first captured clear and complete point cloud images of objects by placing them within the visual range of an RGBD camera, ensuring that the entire visible surface of each object was recorded by rotating the point cloud acquisition device around the target object. This meticulous process allowed us to accurately capture the geometric shapes and surface textures of the objects, facilitating precise data collection and texture extraction.

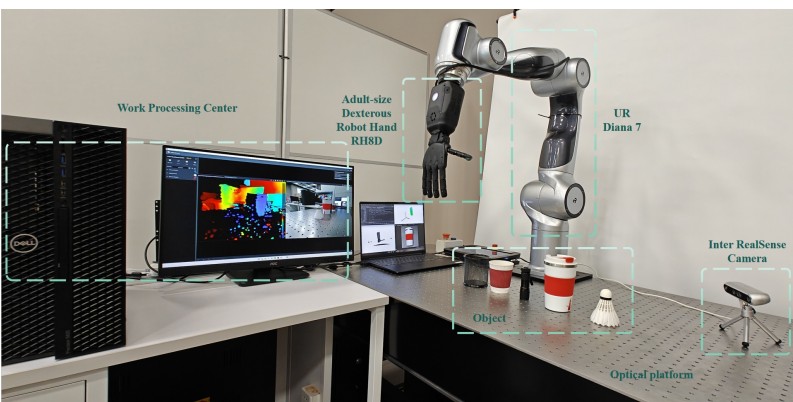

Figure 5: Dexterous manipulation platform with a Seed Robotic Hand, an AGILE ROBOTS Diana 7 arm, an Intel RealSense Camera, an optical platform, and a work processing center.

Next, we used the proposed point cloud processing algorithm to extract texture features accurately. During the dexterous hand grasping phase, we controlled the dexterous hand to perform stable grasping with two, three, four, and five fingers, as demonstrated in Figure. 1, recording the rotation angles of each joint throughout the grasping process. In the simulation phase, we imported the dexterous hand model into the simulation environment and accurately replicated the grasping state of the dexterous hand by following the previously recorded fingertip coordinates and joint angles at different time points. This real-time linkage of each joint's degrees of freedom with actual dexterous hand movements and the simulation platform allowed us to record the spatial coordinates of each grasping contact point in the simulation accurately.

Finally, based on the spatial coordinates of the five fingertip contact points obtained from the simulation, we determined the corresponding spatial coordinates of the dexterous hand fingertip contact points in the real world. This comprehensive approach ensured a high degree of accuracy in replicating and analyzing the dexterous hand's grasping actions.

**Realistic Experimental Scenario:** We developed a production line to simulate industrial environments for gripping tasks. The setup includes AGILE ROBOTS DIANA 7 robotic arms, Intel RealSense cameras for depth sensing (0.11 to 10 meters, with sub-millimeter accuracy), and LIPP-MANN L50pro adjustable lighting (3200K-5600K, 0%-100% brightness). The experimental table is positioned near a wall, with the robotic arm and dexterous hand connected via the Falan. The camera, placed at a 45-degree angle, captures point cloud data from various objects (e.g., cups, bottles, PCB boards, and flashlights). The system runs on a 13th-generation Intel Core i9-13900K processor and an NVIDIA RTX 4090 GPU. The experimental platform, illustrated in Figure 5, accurately simulates industrial conditions, ensuring the reliability and practicality of the research results.

**Texture Feature Point Determination:** Specify the threshold $n_{i,thresh}$ of the Y component of the unit normal vector and the threshold $\sigma_{i,thresh}$ of the variance of the grayscale information, for each point $p_i$, calculate the absolute value of axis $(|n_i|_Y)$ of the Y component of the unit normal vector of it, and if the following conditions are satisfied, it is considered to be a texture feature point:

## 4.2 RESULTS

Our experiments evaluated the accuracy of 3D fingertip contact point localization and pseudo-tactile information extraction, focusing on two key aspects: localization precision and tactile data integration. The system was tested under various finger configurations and object materials to assess its robustness.

- Localization Precision Localization precision was assessed by comparing simulated 3D fingertip coordinates with real-world measurements over 100 trials per object. The root mean square error (RMSE) remained low, with slight variations depending on the number of fingers used, as shown in Table 4, which provides the estimated 3D contact point coordinates for two, three, four, and five-finger grasps.

- Material-Based Performance The system was evaluated across different materials, including glass, plastic, metal, and feather-like textures. As shown in Table A.3, the localization error remained consistent across most materials, with feather-like objects exhibiting the largest deviation due to surface irregularities.

- Pseudo-Tactile Data Integration The system effectively integrated 3D fingertip localization with pseudo-tactile information by matching contact point coordinates with extracted texture features. This demonstrated the framework's capability to simulate tactile feedback based on visual data, without the need for physical sensors.

Overall, the system demonstrated high precision in both localization and pseudo-tactile integration. The average RMSE across 49 objects was 0.92 mm, highlighting the reliability of the approach in various real-world scenarios. These results emphasize the potential of vision-based tactile sensing for robotic manipulation.

## 5 CONCLUSION AND FUTURE WORK

This study introduces a novel vision-based framework for extracting pseudo-tactile information and accurately localizing fingertip contact points during dexterous hand manipulation. Experimental results demonstrate that our approach effectively simulates tactile characteristics using visual data, achieving a localization accuracy of 1 mm and maintaining robustness across varied environmental conditions. The precise localization of fingertip contact points, combined with the corresponding pseudo-tactile information, enhances tactile feedback reliability for robotic grasping, as shown in Figure 6. This integration improves both the dexterous hand's performance and its understanding of object interactions. Future work will focus on increasing the framework's adaptability through dynamic filtering strategies that respond to environmental changes. We also plan to expand the dataset to cover a broader range of object shapes, materials, and lighting conditions, improving the algorithm's generalizability. Additionally, we aim to incorporate multiple sensory inputs, such as additional camera perspectives or tactile feedback systems, to further enrich pseudo-tactile information and enhance the dexterous hand's interaction capabilities. To promote scientific transparency and facilitate reproducibility, we have made our source code and dataset publicly available. This accessibility encourages further exploration and refinement of our approach, paving the way for advancements in robotic grasping technologies.

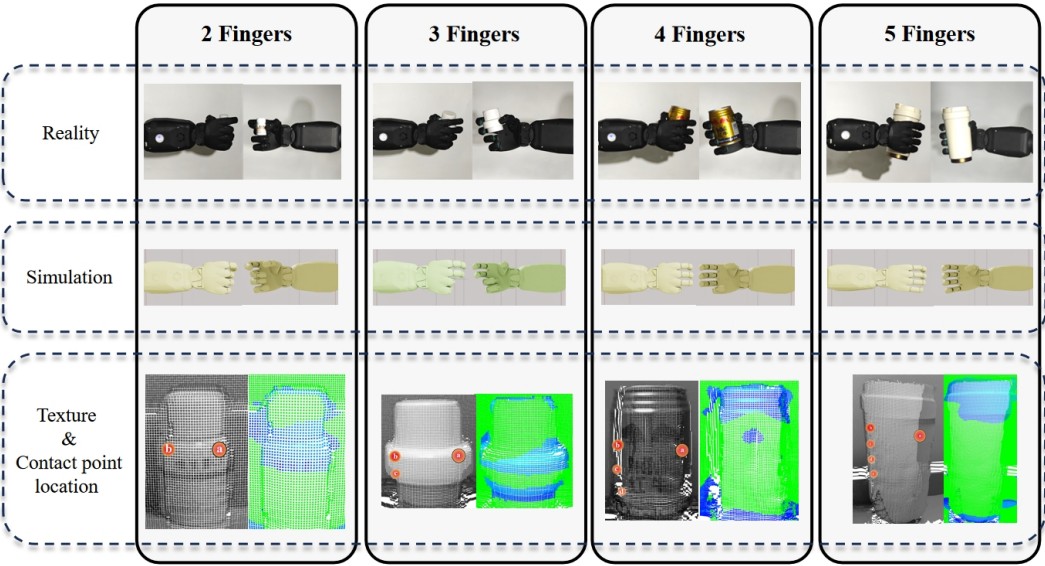

Figure 6: Schematic diagrams of dexterity in grasping objects with different fingers and obtaining surface texture and spatial coordinates of contact points.

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

# A APPENDIX

## A.1 OPEN DATASET AND PRESENTATION OF RESULTS

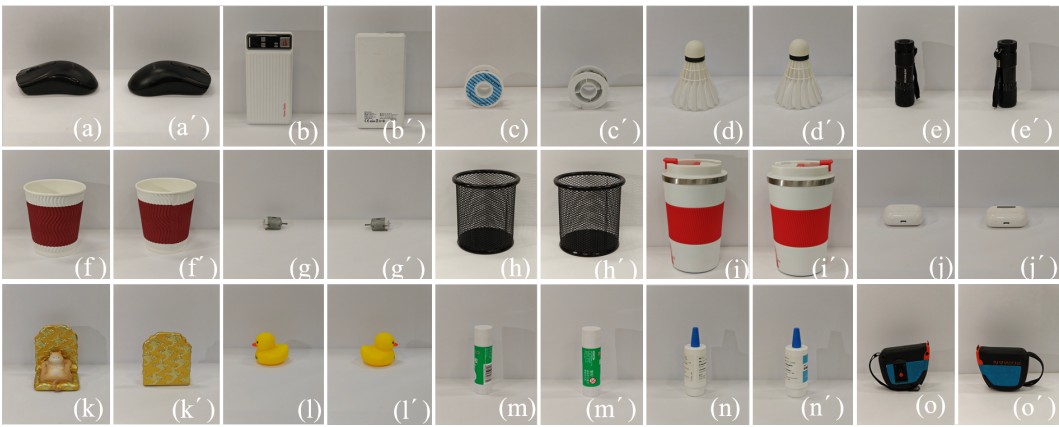

Figure 7: Partial dataset: 15 of 200 common objects with different surface textures and very different sizes.Open data set link: https://drive.google.com/drive/folders/1Pt3kzDJRNTKejL4G8GHUoytgcM9pCAds?usp=sharing.

## A.2 POINT CLOUD EXTRACTION OF TACTILE FEATURE DETAILS

Table 2: 3D Coordinate Data for Each Fingertip (Unit: m)

| Finger Name | Opposable Thumb | Index Finger | Middle Finger | Ring Finger | Little Finger |
|---|---|---|---|---|---|
| **X Coordinate** | 0.0398 | 0.0308 | 0.0074 | -0.0116 | -0.0357 |
| **Y Coordinate** | -0.0694 | -0.0577 | -0.0560 | -0.0577 | -0.0585 |
| **Z Coordinate** | 0.2062 | 0.2757 | 0.2773 | 0.2731 | 0.2681 |

*The coordinates of the contact points of the fingertips of the five fingers of the dexterous hand grasping the object are shown in in Table 2.

Table 3: Parameter Values and Corresponding Hand Index

| Parameter | Cup | Cup# | Can | Can# | Cap | Cap# | Vial | Vial# |
|---|---|---|---|---|---|---|---|---|
| **A1** | 32 | 32 | 32 | 32 | 32 | 32 | 32 | 32 |
| **A2** | 0.05 | 0.05 | 0.05 | 0.05 | 0.05 | 0.05 | 0.05 | 0.05 |
| **A3** | 0.15 | 0.15 | 0.15 | 0.16 | 0.2 | 0.2 | 0.2 | 0.13 |
| **A4** | 0.002 | 0.002 | 0.0015 | 0.002 | 0.01 | 0.01 | 0.002 | 0.0015 |
| **n** | Five | | Four | | Three | | Two | |

*The meanings of the parameters in Table 3 are as follows:

- A1: leaf_size: The minimum number of searches in a KD-tree.
- A2: kdtree_radius: The radius of the neighborhood.
- A3: threshold_normal_y: A threshold for the Y component of the unit normal vector.
- A4: threshold_color_var: A threshold for the variance of the color value (gray value).
- n: Number of fingers used.
- #: Backside of the object.

## A.3 RESULTS OF THE EXPERIMENT

Table 4: Estimated coordinates of contact points in different states (Unit: m)

| | 2 Fingers | 3 Fingers | 4 Fingers | 5 Fingers |
|---|---|---|---|---|
| **A(Thumb) coordinate** | (0.2557,-0.0634,-0.0634) | (0.0316,-0.0714,0.2054) | (0.0313,-0.0675,0.2159) | (0.0201,-0.0658,0.2203) |
| **B(Index) coordinate** | (0.0246,-0.0580,0.2557) | (0.0300,-0.0615,0.2708) | (0.0327,-0.0515,0.2763) | (0.0276,-0.0543,0.2727) |
| **C(Middle) coordinate** | \ | (0.0069,-0.0579,0.2623) | (0.0080,-0.0478,0.2781) | (0.0036,-0.0488,0.2706) |
| **D(Ring) coordinate** | \ | \ | (-0.0128,-0.0485,0.2773) | (-0.0160,-0.0458,0.2671) |
| **E(Small) coordinate** | \ | \ | \ | (-0.0398,-0.0472,0.2613) |

Table 5: Localization error in different gripping situations

| | 2 Fingers | 3 Fingers | 4 Fingers | 5 Fingers |
|---|---|---|---|---|
| **Localization Error (m)** | 0.0007 | 0.0009 | 0.0011 | 0.0012 |

Table 6: Localization error in different gripping material

| | Glass | Plastic | Metal | Feather |
|---|---|---|---|---|
| **Localization Error (m)** | 0.0008 | 0.0009 | 0.0006 | 0.0010 |

