# OpenReview forum: "Vision-Based Pseudo-Tactile Information Extraction and Localization for Dexterous Grasping"
_ICLR.cc/2025/Conference — Submitted to ICLR 2025_

### Official Review · Reviewer_bzE2 · 2024-10-29

**Soundness:** 2
**Presentation:** 2
**Contribution:** 1
**Rating:** 3
**Confidence:** 3

**Summary:**

This paper investigates the challenges of acquiring haptic perception during grasping by a mechanical dexterous hand and proposes solutions. The main tasks of the research include the acquisition of “pseudo-tactile” information about everyday objects through vision and the construction of a dexterous hand (RH8D) model in Isaac Sim for real-time fingertip contact localization. The study establishes a scientific link between simulated 3D coordinates, actual 3D coordinates, and pseudo-tactile information derived from the point cloud, which is quantified by normal vector and grayscale ANOVA. Experimental results show that the method is able to clearly extract the surface texture of an object, accurately locate the fingertip contact point in real time, and provide haptic information at the contact point.

**Strengths:**

This paper addresses the problem of obtaining tactile information during grasping based on only vision perception, and provides a clear method for object surface texture extraction with 3D point cloud input. The strengths are listed below.
1. The authors provide abundunt and clear explanation for method presentation, and present a simple yet effective approach for point cloud preprocessing and feature extraction.
2. The authors give a thorough representation on the expermental setup, and conduct real-world experiments on a dexterous hand for validation.
3. The result of the experiments seems very ideal, indicating the effectiveness of the proposed method.

**Weaknesses:**

While I recognize some of the article's contributions, I still have the following concerns.
1. From my point of view, this paper is a bit lack of novelty, because some parts of the method chapter are biased towards engineering practice rather than innovation, such as point cloud preprocessing, camera coordinate system transformations, and other types of work are actually common in robotics work, and cannot be listed as points of innovation. Meanwhile, the texture extraction method in the section overlaps most with [1].
2. While generating pseudo-haptic sensing is one of the important contributions of the article, I didn't see that the authors had measured how good the quality of the generated haptic signals were, both quantitatively and qualitatively.
3. The article's experiments still seem inadequate to me and lack comparison with previous work. For the contact position localization part, are there any previous baselines that can realiza this? For example, 3D point cloud keypoint prediction baselines, etc. Meanwhile, the authors didn't conduct the ablation studies on the proposed method, such as the different effectiveness on selections of KDTree radius, normal threshold, etc.

[1] Budiyanta, N. E., Yuniarno, E. M., & Purnomo, M. H. (2021, December). Human point cloud data segmentation based on normal vector estimation using pca-svd approaches for elderly activity daily living detection. In TENCON 2021-2021 IEEE Region 10 Conference (TENCON) (pp. 632-636). IEEE.

**Questions:**

Apart from the weaknesses I have mentioned, I still have some questions for the authors to respond.
1. I noticed that the author evaluated the localization results by "comparing simulated 3D fingertip coordinates with real-world measurements", how to acquire accurate real-world contact positions with 3D point cloud input or other approach?
2. The authors use Intel RealSense Camera for capturing depth images and convert them into point clouds. As far as I know, depth images generated by D435 can be very noisy, thus the estimation of surface normal vectors can be very biased, how to deal with this problem?
3. the authors mentioned that objects with glass material is included in the dataset, but the depth images generated from RealSense Camera on transparent objects are catastrophy. How can the author obtain accurate point cloud information under this circumstances?
4. In Sec. 3.3, the author mentioned "We also employ policy fine-tuning techniques, using a small amount of real-world grasping attempt data to fine-tune the model", but since the method is not deep-learning based, how to conduct the fine-tuning stage in the pipeline?

Right now, I'm inclined to reject this paper. However, if the authors are willing to answer my questions well, I'll consider raising my score.

---

### Official Review · Reviewer_ZwPV · 2024-11-02

**Soundness:** 2
**Presentation:** 2
**Contribution:** 1
**Rating:** 3
**Confidence:** 3

**Summary:**

This work presented an approach to acquire object surface information including textures and geometries, and proposed a simulation-involved approach to locate fingertips of robotic hand for grasping. The results demonstrated the object grasping with different number of fingers and object surface feature extraction.

**Strengths:**

The work has clear writing and good structure which makes it easy to follow and understand. Figures are well-made and informative. It is well-motivated to address the hard-to-acquire tactile perception by using vision for dexterous robotic hands.

**Weaknesses:**

I do not think this work has solid contribution or concrete experimental results. The proposed method simply combining existing techniques such as extracting point cloud from RGBD camera, and using simulator to simulate grasping. Instead of quasi-static grasping, I would encourage authors to extend it to more dynamic manipulation tasks and explore whether the extracted object features can be leveraged for these complicated tasks.

**Questions:**

There lacks of the definitions of the "grayscale variance" and "Y-component", the authors could define them properly when they first appear.

Table 1 and equations (1)-(4) are describing the transformation from camera frame to the world frame to extract point cloud which seems to be fundamental computer vision techniques. I would suggest to move them to the Appendix since it is not novel in this work.

---

### Official Review · Reviewer_Jrkz · 2024-11-02

**Soundness:** 1
**Presentation:** 1
**Contribution:** 1
**Rating:** 1
**Confidence:** 4

**Summary:**

This paper presents a dataset of point cloud images of a wide variety of everyday objects. A dexterous hand model is built in an Isaac Sim simulator for real-time contact localization by replicating the real-life set up in simulation and using measured joint angles.

**Strengths:**

Open-sourced dataset of point cloud images is made available.

**Weaknesses:**

Details are so unclear it is difficult to fully understand the paper. For instance,
- Paper repeatedly talks about the "Y component of the normal vector" without clearly defining a coordinate frame. Authors mention employing "policy fine-tuning techniques" on page 7 without ever mentioning a policy up until this point.
- The role of simulation in the paper is not clear. The paper mentions "This real-time linkage of each joint’s degrees of freedom with actual dexterous hand movements and the simulation platform allowed us to record the spatial coordinates of each grasping contact point in the simulation accurately." This sounds like a simple forward kinematics problem that would only require a mathematical model of the robot – not a full-fledged simulation.
- Paper claims that intel realsense has “sub-mm accuracy”. This is not supported by documentation from the manufacturer: https://dev.intelrealsense.com/docs/tuning-depth-cameras-for-best-performance?_ga=2.110331777.520332705.1730517789-101245430.1730517789#section-verify-performance-regularly-on-a-flat-wall-or-target
- Section 4.2 claims to assess localization precision. It is unclear what the ground truth is, how this is being measured and what measurement is being compared against this ground truth. Referenced Table 4 is difficult to understand, has no mention of errors or comparative ground truth.
- An RMSE error is reported with no clarity on what quantities are being compared.
- It is also unclear how related work Section 2.3 is connected to this work. Pseudo-haptics is associated with giving humans touch sensory feedback, whereas “pseudo-tactile” in this paper is related to analyzing the surface tactile properties of objects.

Minor comment:
Paper is very poorly formatted. Main result tables are placed in the appendix and are difficult to understand. Results in Tables 4,5 and 6 have poor choice of units (meters when dealing with textures that are likely sub-cm scale), and numbers are represented with arbitrary precision with no regard for the precision and error rates of the depth measurement device, ie. Intel Realsense cameras. Bullet points in the results section seem to have headers that have the exact font formatting as the rest of the text.

**Questions:**

- What is the purpose of using simulation?
- How is the tactile information from the point cloud used in contact localization?
- How is ground truth obtained for contact localization? What is the precision/resolution of the ground truth measurement?
- What does policy finetuning have to do with this paper?
- How are the texture feature points used once they're determined by thresholding?

---

### Official Review · Reviewer_gCsm · 2024-11-04

**Soundness:** 1
**Presentation:** 2
**Contribution:** 1
**Rating:** 3
**Confidence:** 4

**Summary:**

This paper presents a framework to acquire point cloud representations of objects and simulate the contact locations when using a dexterous robotic hand for grasping tasks. The point cloud data are first processed to filter the background and texture feature points are determined afterwards. The real grasping data are derived from the hardware and can be reproduced in a simulation environment with the corresponding contact locations and texture features.

**Strengths:**

This work introduces an approach to extract pseudo-tactile information from vision and contact locations for robotic grasping tasks. The topic is interesting, the pipeline is presented with details, and the experiment results show the effectiveness of the approach with high accuracy.

**Weaknesses:**

The overall contribution is marginal. The vision-based "pseudo tactile" features are derived using some common tools for point cloud data processing. The simulation for finger tip localization is from replicating and transforming the real motion data into the simulated environment. It is also claimed that the combination of this vision-based information and the fingertip contact points can enhance tactile feedback reliability in robotic grasping. However, this is not clear. The experiment does not show how the robotic grasping is improved.

**Questions:**

1. In the experiment, the motion of the real robotic hand is recorded and then replayed in the simulation to show the contact locations can accurately match. I am not clear the motivation here.
2. In the texture feature extraction framework, how the conditions to classify points as texture features work?
3. In pseudo-tactile data integration part, it is shown that the system is able to simulate tactile feedback without the need for real sensors. How the feedback works? Is the pseudo-tactile information used for some closed-loop controller?

---

### Meta-Review · Area_Chair_1rPA · 2024-12-19

**Metareview:**

This paper introduces an approach for extracting pseudo-tactile information for robotic grasping and integrating a simulated dexterous hand model. The work was evaluated by the reviewers as lacking in novelty and depth, as it primarily combines existing techniques without significant innovation. The experimental validation was limited, with unclear metrics, insufficient baselines, and unexplored claims regarding the effectiveness of the proposed method. Additionally, the  reviewers have pointed that the presentation required significant improvement, as there were a lot of ambiguities in used terms, inconsistent reporting of results, and unsupported hardware experimentation claims.

**Additional Comments On Reviewer Discussion:**

The authors did not participate in the rebuttal phase, leaving critical concerns raised by the reviewers unresolved.

---

### Decision · Program_Chairs · 2025-01-22

Reject